# Impact of the COVID-19 Pandemic on CSR Activities of Healthcare Providers

**DOI:** 10.3390/ijerph20010368

**Published:** 2022-12-26

**Authors:** Christina Deselaers, Alina Dahmen, Sonia Lippke

**Affiliations:** 1School of Business, Social and Decision Science, Constructor University (Formerly Known as Jacobs University Bremen gGmbH), Campus Ring 1, 28759 Bremen, Germany; 2Klinikum Wolfsburg, 38440 Wolfsburg, Germany

**Keywords:** healthcare professionals, COVID-19, CSR, organization, experience, leadership, crisis management, strategic management, corporate social performance, crisis communication

## Abstract

(1) Background: Corporate social responsibility (CSR) is important for every company that cares for sustainable structures. Healthcare providers especially have made social responsibility their goal. However, crises such as the COVID-19 pandemic impacted different activities within the healthcare sector including CSR and its monitoring. However, theory-driven CSR research within the healthcare sector is scarce and monitoring requires a structured understanding of the processes. Therefore, the objective of this study was to investigate the CSR practices and activities which healthcare providers have implemented in an exemplified country namely Germany and the effect of the pandemic in this process. (2) Methods: Participants were sampled based on their field of care (general, psychiatric, or rehabilitation), the type of organization (public, private, or non-profit), and group membership. A total of 18 healthcare providers were initially recruited, out of which nine participated in the interviews. They represent companies with yearly revenue of between EUR 110 million and EUR 6 billion, and have between 900 and 73,000 employees. (3) Results: CSR-related activities were postponed due to times of crisis. There was a necessity to rapidly digitalize processes. Frequent and precise communication turned out to be important for keeping employees’ well-being, motivation, and satisfaction levels high. Environmental efforts were counteracted by new hygienic requirements and a shift in priorities. Many study participants expressed the hope that after the pandemic, newly established methods, processes, and structures (e.g., digital meetings, quicker and more inclusive communication) would be maintained and developed further. (4) Conclusions: The pandemic has been challenging and at the same time, these challenges also created opportunities to strike a new path using the learnings to overcome future health-related or economic crises.

## 1. Introduction

Climate changes have a strong impact on the health of individuals and companies. Thus, three aspects are immensely important: (1) having strategies contributing to buffer the overall climate change; (2) deriving action plans for moderating the effects of climate challenges on individuals and companies; (3) and most importantly, working to protect the climate through adequate company behavior. Exchanging and communicating different perspectives and knowledge to tackle the challenges can be done through corporate social responsibility (CSR).

### 1.1. The Definition of CSR

Definitions of the topic strongly vary and so do the terms frequently used in this context. Apart from CSR, “corporate sustainability” [1] or “triple bottom line” [2] are often utilized. All previous researchers seem to agree that the following elements need to be included in a definition of the concept:I.The three main areas of engagement are society, environment, and business—whereby society usually refers to doing something good for society; involvement in environmental activities ranges from supporting environmental and climate initiatives by only sourcing local or fair-trade products, to re-evaluating and changing a company’s statutes to become more (i.e., energy) efficient; and business denotes economic decisions, such as becoming an attractive employer through extending additional benefits to employees or by making processes more efficient, all leading to increased satisfaction and sustainability. Quite often, approaches to support and strengthen human rights or to take the opinions of customers into account explicitly mention these three elements to emphasize their importance [3,4,5]. Complementing these are the legal and ethical spheres [6], elements of which are found in all three areas. In addition, fundamental philanthropical ideas can often be found in various CSR activities [7].II.External social, economic, and environmental factors play into the decision-making processes within a company. Such factors can include, for example, regulators, employees, shareholders, customers, activists, researchers, or any other form of stakeholder [8].III.Stakeholders legitimize the company’s CSR activities. This is often shown in terms of how much support a company gets for its activities, i.e., donations, time employees maintain in the company before they turn over or spend on helping out and engaging [9,10], and how much stakeholders trust that the company’s image is befitting of its actions [11,12].

CSR is rooted deeply within business practices and applied activities: companies have always tried to improve their image or create a competitive advantage to generate more business, get more clients, or recruit and retain the best employees [3,4,8,13,14,15]. Those CSR-related activities are actions and outcome experiences that generally include social, economic, and ecological practices that impact the corporation’s reputation, organizational development, and engagement of stakeholders [8,16,17]. 

Social challenges and climate issues are frequently factored into economic decision-making since a company’s presentation and stance within society shape its attractiveness [15,18]. Thinking responsibly about organizational resources—i.e., human, natural, or artificial—and using them sustainably creates a strong link between corporate social performance and psychological and economic rationales. As CSR can also affect attractiveness and competitiveness, thus it functions as a new type of leadership [19]. According to Bauman and Skita [13], this behavior fosters innovation, expands the horizon, and rewards those who help create value for the organization.

### 1.2. Corporate Social Responsibility—Concepts, Models, and Theories

With CSR technique, health risks relating to climate change can be communicated, and effective strategies to mitigate and adapt to changes can be applied. These are especially essential for audiences that might not be informed or who have historically tended to resist this information. The focus on CSR activities is steadily increasing as its versatility is regarded as beneficial to organizations. CSR activities are actions and outcome experiences to improve the company’s image or to achieve a competitive advantage by building trust, appealing to consumers, and persuading internal partners that the company is fulfilling societal requirements. 

CSR activities influence a corporation’s reputation and the engagement of stakeholders, defining organizational development [8,17,20]. Social challenges, health-related aspects, and climate issues are frequently factored into economic decision-making since a company’s presentation and stance within society shape its attractiveness [18]. Thinking responsibly about organizational resources, i.e., human, natural, or artificial resources, is often strongly connected to economic and psychological rationales and a step towards a new kind of leadership [19]. This new way of thinking broadens horizons, fosters innovation, creates new production schedules, and rewards the people who help create value for the company [13].

Corporate social responsibility (CSR) is a concept that is based on a company’s will to go beyond its obligations to maximize social, environmental (with a strong focus on climate), economic, and ethical welfare [19,20,21,22,23,24]. Throughout the last decade, the concept was brought into the focus of large corporations, activist groups, and political organizations. Its roots, however, stretch back to the 1920s when Clark first talked about the ‘social control of businesses’ [25]. On a larger scale, CSR in Europe is defined by the standards published by the European Union (EU) and incorporated in the International Organization for Standardization (ISO) norm 26000. The EU simply says CSR is “the responsibility of enterprises for their impact on society” [26]. 

Similarly, the CSR definition of ISO 26000 entails that businesses have to operate socially responsibly. This requires transparent and ethical behavior, and clear communication contributing to the overall well-being of society including disadvantaged persons [27]. In the context of this research, “society” also touches upon the business realm. Employees are part of society at large and enterprises shape societies by creating jobs, fostering likes or dislikes, and interfering with climate. In addition, the United Nations have an ambitious global portfolio of the so-called 17 sustainable development goals (SDGs) [28], consisting of core aspects to tackle climate challenges, create (more) equality amongst people, and make businesses more sustainable.

### 1.3. Corporate Social Performance and Social Responsibility Theories

Various models and theories shape CSR research. The following section looks at the most prominent ones, without any claim to providing a complete overview of all existing concepts and ideas. The corporate social performance model for businesses was proposed by Carroll in the year 1979 as an answer to the highly diverse discourse about corporate social responsibility and its elements. The author suggests that three aspects need to be considered, namely “the definition of social responsibility, the social issues involved, and the philosophy of responsiveness” [29] (pp. 499–502).

For his three-dimensional conceptual model, social responsibility includes four realms: first the economic realm, because a business’s most fundamental element is its economic endeavor of producing services and goods for society. The second element is the legal sphere defining the rules by which businesses are expected to conduct their activities. Ethical responsibilities are the third, stressing society’s emphasis on a specific behavior that is not manifested by law. Lastly, the so-called “discretionary responsibilities” go even further than the three others, although their definition is not clear-cut and often subject to individual judgment and interpretation. “This four-part framework can thus be used to help identify the reasons for business actions as well as to call attention to the ethical and discretionary considerations that are sometimes forgotten by managers” [29] (p. 501).

The involved social issues are far more difficult to define exhaustively as they change over time or vary among situations. To outline the overall model, Carroll identified “consumerism, environment, discrimination, product safety, occupational safety, and shareholders” as six elements commonly considered. However, each business needs to develop them individually. Lastly, a business’s social responsiveness is part of the model. It can range from proactively doing a lot to not responding at all. Again, each manager needs to set the scope for actions based on the company’s situation. For creating the model, Carroll specified the categories as “reaction, defense, accommodation, and proaction”.

Overall, the model is designed for managers and academics. For managers, the model is a tool to develop an understanding of how economic performance and social responsibility are joined together. Academics on the other hand get an encompassing model that combines various definitions of social responsibility and shows that they are systematically interlinked and not separate from one another.

### 1.4. The Pyramid of CSR by Carroll

The pyramid of CSR, introduced by Carroll in the year 1991, has four different levels, signifying the economic (profitability), legal (within the provided legal framework), ethical (fair and appropriate actions), and philanthropic (active engagement in socially beneficial activities) responsibilities [5]. The basis for all is that society gives directives according to the four levels, which are then considered to be binding for a company. The defined economic responsibilities include the provision of goods and services for society, while profits are the “primary incentive for entrepreneurship” [5] (p. 40). 

The idea for companies to make some profits, to be economically stable, was transformed into the notion of profit maximization [5] (p. 41), which still represents today’s most commonly exercised objective. Apart from the expectation of profits, companies are also supposed to operate within given legal boundaries. “As a partial fulfillment of the “societal contract” between business and society, firms are expected to pursue their economic missions within the framework of the law”, is how Carroll [5] (p. 41) phrases the definition. 

Societal members look even further than the purely economic and legal responsibilities of a company. Although both levels are based on certain ethical understandings, creating an individual level, namely the third, for these concerns, highlights their importance. Carroll sums it up as follows: “Ethical responsibility embodies those standards, norms, or expectations that reflect a concern for what consumers, employees, shareholders, and the community regard as fair, just, or in keeping with the respect or protection of stakeholders’ moral rights.” (p. 41). The highest and fourth level then focuses on the expectation that corporate actions should show good corporate citizenship, emphasizing the promotion of “human welfare or goodwill” [5] (p. 42).

Other scholars, especially Wenzel argues that societal expectations play a too large role in this model [30], especially considering that societies consist of various groups whose interests are not as easily combined as suggested in the model. In addition, other scholars argue that the positioning of the four responsibility spheres within the pyramid works well for corporations in Western countries, but is very different, for example, in Africa because priorities are naturally different [23,24,25,26,27,28,29,30,31,32,33]. Carroll acknowledged in the year 2016 that CSR is not a concept set in stone but is flexible and influenced by the time scholars look at it [34]. Therefore, he outlines that his framework and other existing ones might change over time, but the essence of acting responsibly on different levels will always play a role in CSR research.

### 1.5. The CSR Model by Quazi and O’Brian

Quazi and O’Brian’s model from the year 2000, divides CSR activities along a horizontal and a vertical axis [7]. The horizontal axis focuses on the so-called “wide” long-term and “narrow” short-term responsibilities. On the vertical axis, the authors divide activities between their costs and benefits. 

The four created quadrants are very similar to Carroll’s four levels and describe the modern view, the social economic view, as the two elements that are considered to benefit from CSR, and the philanthropic view and the classic view, rather linked to costs of CSR (p. 36). Wenzel also criticizes this model for rather looking at individual actions than the entire responsibility a company has for society [30].

### 1.6. The Three-Dimensional CSR Approach by Carroll and Schwartz

In the year 2003, Carroll and Schwartz introduced a slightly changed model for CSR (compared to Carroll’s 1991 proposition [5]), called the three-dimensional approach [6]. This new model does not include the philanthropic level any longer. Consequently, the previous hierarchical structure of the other three was modified into a flexible one, allowing for overlaps between the different dimensions, creating altogether even categories:(i)Only economic(ii)Only legal(iii)Only ethical(iv)Combined economic-legal(v)Combined economic-ethical(vi)Combined legal ethical(vii)Combined economic-legal-ethical

Especially the first three points are often the starting point for critical remarks about the model, as the authors mention themselves and as others emphasize in their work, respectively [30,35]. The reasoning behind the criticisms is that having a pure category—either economic, legal or ethical—is not quite possible, because of their interdependency. 

Actions in one field will almost always be in some form of correlation with another. The most commonly noted example is “businesses have a moral obligation to respect legitimate laws” [36] (p. 226), therefore all economic actions are influenced by legal matters. Some scholars, such as Solomon, very early on, raised the issue of morals that should be reflected in the ethical domain, but are not included by the authors [37]. As with all models, there are many critical voices, but generally, the three-dimensional model is able to sketch, in a simple manner, the complexity of CSR.

Many theorists, such as Donaldson and Preston [38] and Costa et al. [16] for example, argue that a manager’s role is to satisfy all groups of people (stakeholders) that can influence the work of a company. Such influencing stakeholders may be contractors, customers, employees, investors, media, shareholders, etc. These stakeholders can either impact the overall opinion about a company positively or negatively. However, Hommel and Godard [39] voicing negative options or even threats is usually more likely than emphasizing a company’s positive actions [40]. 

Shareholders, for instance, “[…] hold a major stand with full legitimacy to ask, in addition to fiduciary duties, the firm they own to engage in CSR” [8] (p. 120). If investors look at the value base and communication of action of companies, they consider giving money to institutional ownership [41,42,43,44,45], and image creation [21,46] play an important role. The expectations of the stakeholders are not set in stone, but change regularly, depending on current organizational decisions and potentially also media coverage of certain topics, e.g., environmental issues [47].

### 1.7. Corporate Social Irresponsibility

Unfortunately, not all individuals and organizations act socially responsible, and therefore the concept of corporate social irresponsibility (CSI) shall not go unmentioned. In the year 1977 an article was published defining CSI as follows: “A socially irresponsible act is a decision to accept an alternative that is thought by the decision maker to be inferior to another alternative when the effects upon all parties are considered. Generally, this involves a gain by one party at the expense of the total system.” [48] (p. 185). While corporations actively engaging in CSR, use their resources and influence to do good, those on the edge of CSI are aware that the respective actions can be extremely dangerous for a corporation, society, or the environment at large [49].

While various reporting standards for CSR practices have been introduced in the last couple of years (e.g., ISO 26000, Deutsche Nachhaltigkeitskodex, etc.), few systematic publications exist on standardized CSR monitoring, and no specific theories have been developed on the topic so far. Nonetheless, such standardization is required to compare various CSR activities and assess their efficiency.

### 1.8. The Development of CSR—A Scoping Review

Apart from the theoretical background information on CSR, analyzing the existing literature based on the monitoring contribution provides an even broader base for the following analysis. Therefore, a scoping review was conducted with a Scopus search, providing a framework for identifying eligible records to be included in the analysis [50]. Scopus, a superior search engine for scientific works, was chosen based on the depth and breadth of materials available. The keyword combination ’corporate social responsibility’ AND ‘monitoring’ yielded literature of interest pertaining to the topic of this review. Although the review is very thorough, it does neither claim to be fully encompassing nor to be inclusive of all existing materials [50].

Due to the volume of records which were found in the search, the documents were analyzed in a two-step process. The first step was a comprehensive screening of the papers’ abstracts. The second step involved an in-depth analysis of the content of the remaining records identified through the abstract screening process. Therefore, when abiding by strict rules and consistently applying the same inclusion and exclusion criteria, this method can be utilized and was hence chosen [50]. To decide if an abstract was to be excluded, the following questions had to be answered with “yes” (for inclusion), “no” (for exclusion), or “unsure”:(i)Is corporate social responsibility mentioned in the abstract and in the keywords?(ii)Do the authors refer to any form of monitoring of activities related to corporate social responsibility?(iii)Is the sample size large enough to be statistically significant?

All abstracts that were labeled “unsure” were reevaluated. The procedure was repeated and those not immediately excluded were considered for the second phase of the analysis. This ensured that no information was left out solely because the abstract did not yield a clear-cut picture of the manuscript’s content. Therefore, all qualifying documents (“yes”/“unsure”) were further processed in the second step, the in-depth analysis of the manuscript. Here, the focus was not on the abstract, but on the full text to gather information on CSR monitoring in as much detail as possible [50]. Accordingly, the manuscripts that did not yield any information involving the research topic were excluded. The applied final exclusion factors were defined a priori:(i)the full text was not accessible, and the authors did not respond to inquiries within 6 weeks;(ii)the content was based on a very narrow or a too specific study sample (e.g., one company or one country plus one specific sector), so broader applications of the input did not seem possible;(iii)CSR was mentioned, but it was not the focus of the paper;(iv)the CSR focus was purely on improving human rights or the situations of workers;(v)CSR was only mentioned in the combination of reporting certain corporate activities, but monitoring did not play a part.

This systematic review is composed of 486 manuscripts (search in March 2020 with *N* = 327, an updated search in September 2020 and July 2022 with *N*_additional_2020_ = 15 and *N*_additional_2022_ = 144) of which 350 are journal articles and the remaining 136 are a mix of books or chapters thereof, conference papers or reviews, short surveys, notes, and editorials. Of the manuscripts, 471 were already published and 15 were in press. The titles were published between 1987 and 2022 in various fields, such as medicine, social or environmental/climate science, business, finance, or management.

Based on the screening of abstracts, 75 papers were analyzed in-depth. In the updated search, 8 additional manuscripts were included that are suitable for the systematic review. All other manuscripts were ruled out due to a lack of specific information on CSR monitoring (compare Figure 1).

While the initial focus was set on how CSR activities can be monitored independently from monitoring certain individuals or stakeholders, it became evident that very little input was available on this topic. However, another aspect of monitoring, namely the performance of managers and their communication styles, has been under more scrutiny than anticipated. Thus, it is also included in the systematic discussion of results, because this element of monitoring creates an understanding of how strong psychological components (such as trust or control) interrelate with business decisions.

The included records were analyzed based on the input they provided on CSR monitoring. The content was sorted by the following topics:(i)Common understanding—general information(ii)The convergence of standards(iii)Monitoring bodies(iv)Monitoring mechanisms(v)Acknowledging influence(vi)Monitoring costs to give an overview(vii)(Shareholders) monitoring of individual people (i.e., mostly managers)

Even though the sample size was comparatively large, the manuscripts generally did not provide deeper insights into how CSR was monitored. The information on and scientific analysis of the topic is very limited in both scope and quantity. However, the search yielded results regarding the importance of different monitoring bodies, the CSR standards that could be the basis for developing monitoring instruments, and the costs of monitoring, always relating to communication.

Overall, this systematic review shows that previous studies on monitoring corporate social responsibility activities have focused on seven different topics. Some (*N* = 3) start with a rather general description of CSR connecting it ever so slightly to monitoring. Other scholars (*N* = 5) look at the existing CSR standards and theories (e.g., stakeholder theory, the pyramid of CSR, or the three-dimensional approach to CSR) and try to combine them. Though only limited references to monitoring are made here. Six manuscripts deal with different institutions that take over the monitoring and three have references to specific monitoring mechanisms. Other manuscripts (*N* = 4) take into consideration who can influence monitoring systems. The costs of monitoring are only considered in two manuscripts and the specifics of monitoring managers are covered in three. Important topics to be analyzed seem to be who monitors whom (i.e., shareholders monitoring senior managers), the reasons why individuals are monitored (i.e., to ensure appropriate investments), and why external monitoring bodies (e.g., financial analysts) play an important role for shareholders assessment of the quality of their investments (see Table 1).

The results show that in the scientific literature CSR activities are only focused on in combination with another topic. The most pressing combinations with CSR activities seem to be financial or image related, usually focusing on one topic, for example, climate change or health issues. Either stakeholders want to ensure their money is well-spent or a company uses CSR to make a good impression that promotes their business. The links are often analyzed between a set of CSR activities and a corporation’s or manager’s individual performance i.e. having an influence on how meaningful the perceived results are. However, the primary focus is not on how to monitor the activities in detail and by what means CSR is done. These desiderata are accordingly addressed in the following section with an empirical investigation.

### 1.9. Crises-Related CSR in the Healthcare Sector

CSR and its ramifications in the German healthcare sector are not yet systematically researched. This sector underlies very strict federal- and state-related regulations that do not include a specific request for sustainable practices counting as CSR activities. Results of studies from other countries, however, show the benefits of such activities in the healthcare sector: some argue very generally that the concepts of corporate governance and corporate social responsibility provide key managerial elements that can be beneficially adapted to the needs of the healthcare sector [51]. Others more specifically found “that CSR engagement of a hospital at the level of the employee helps to enhance the innovative behavior of employees at the workplace” [52] (p. 10).

Generally, CSR activities are undergoing a crisis-related stress test: Crises such as climate change or the Coronavirus disease 2019 (COVID-19) pandemic impact these business practices by reallocating financial resources and diverting the focus to other (more) pressing topics. So far, the literature on CSR in times of crisis has two main foci, first financial crises [53,54,55] and second organizational (predominantly image) crises [56,57]. The research that singles out the environmental aspects of CSR during crises is extremely limited [58], most research tries to establish an encompassing CSR approach including societal, organizational, and environmental aspects. 

One major crisis in recent years was evidently the COVID-19 pandemic: It has been a strain on healthcare systems around the world. The discrepancy between the continuous provision of essential services, the adaptation to the challenges of resolving the pandemic, and the issues of governance during the pandemic are immense [59,60,61,62]. While many different sectors were affected by the COVID-19 pandemic, the healthcare sector was especially impacted because of the increased workload due to more patients and more required hygiene measures for patients, staff, and visitors [63,64]. “Due to the current outbreak of COVID-19, both public and private sector organizations are facing tremendous functional challenges” [65] (p. 1240) concerning changes in structures, processes, and workforce sustainability. The current circumstances force organizations to “alter their workforce in technical, physical, and socio-psychological ways not seen before” [66] (p. 183), posing great obstacles to the standard way of doing business. The changes are extensive and call for concurring strategic management and leadership as well as crisis management and crisis communication [67,68,69,70,71,72,73,74,75,76,77,78,79]. This paper aims to review the crises of the COVID-19 pandemic empirically and apply a theoretical backdrop to highlight the connection and relationship to CSR. 

Apart from the COVID pandemic, climate change is a relevant factor that negatively influences the health of individuals and societies [61,70,71]. Global warming is one aspect that strongly impacts people all around the world [72]. Having many pointing the finger at companies rather than individuals for having a negative influence on the environment and society, organizations all over the world implement corporate social responsibility or sustainability practices within their entities. In Germany, one of the sectors with no specific regulations regarding CSR is the healthcare sector. Healthcare providers are not required by law to be sustainable although hospitals especially consume and waste an enormous amount of energy because providers very often do not evaluate savings potential [73]. The pandemic has a major impact on environmental efforts, the next few years will show the specific fallouts. Accordingly, this study enriches the research by giving first insights into the effects of the pandemic on environmental CSR activities, building on previous research.

In the field of environmental CSR, research has already reported interesting general aspects: For example, well-written and detailed CSR disclosure contributes to transparency by displaying the environmental and social impact precisely [74]. Fully committing to environmental corporate social responsibility business practices “could avoid public skepticism that would negate the organization’s effort in strengthening customer loyalty” [75] (p. 710). Babiak and Terndafilova looked at sports organizations’ motives to adopt environmental practices and found that “environmental practices in professional sport are driven by two important considerations—the desire to achieve legitimacy and the strategic or competitive advantages that these types of activities might provide.” [76] (p. 22). Additionally, with a specific focus on the healthcare sector, a study by Xu et al. looked at the pro-environmental behavior of healthcare staff connected to their institutions’ attitudes and actions regarding energy efficiency [77]. They found out that the staff learns from what they experience within their institution, and their attitudes and behaviors are thus highly relevant. Employees consequently play an important role and their engagement is one of this study’s main foci.

Previous research showed that certain groups are more vulnerable to the effects of the COVID-19 pandemic: race, gender, living conditions, work environment, and access to healthcare are all determinants influenced by socio-economic status. Results of the first studies conducted during the pandemic have already shown that isolation and discrimination result in anxiety [78] and so does work-related stress [79]. As much as technological advances played an important role during the pandemic, technology was also found to be a major stressor for employees working from home [67].

Empowering and supporting employees to develop positive mental health in their work environment becomes increasingly important [80]. For instance, nursing students need to be taught self-care practices to be equipped to handle crises such as the COVID-19 pandemic [81]. Psychological safety plays a key role in this, and CSR can buffer crisis effects on the individual [82]. Around 60 percent of healthcare professionals worldwide are nurses, namely about 28 million [83]. Healthcare professionals, especially the nursing staff, generally have a high workload. A pandemic such as COVID-19 functions as another stressor, which increases the fear of becoming infected or infecting someone else, not to mention higher exhaustion—physically and mentally—, more difficulties to balance work life and family life, and organizing childcare [66,84,85,86,87].

In many countries and sectors, work–life changed drastically in the year 2020 when infections increased, because employees were either asked to work remotely, went into short-time work, i.e., could only work limited hours, or even became unemployed [88]. In 2016, a study was published by Kornich and Eger [89], explaining that living in difficult economic situations has a strong influence on the decrease in satisfaction with family life. “Ensuring worker protection should be the responsibility not only of citizens but also of organizations” [90] (pp. 39–40). Therefore, it is especially important for employers to critically evaluate their employees’ situations to ensure that those with a higher risk for serious cases are appropriately protected from the virus [62]. Employers should do so in a strategic way and CSR provides such options. Moreover, CSR and communication are important components for retaining employees or attracting new employees, which is especially important in times when skilled labor is in short supply [67,68]. Solutions have hardly been looked at within the scope of CSR. The current study will examine this accordingly and the rationale will be outlined in the following.

It was stated that “The pandemic[‘s] influence on occupational satisfaction seems to be negative” [91] (p. 4) for nurses in the US according to data from 2020 comparing the situation before the pandemic and during the pandemic [92]. A similar trend was reported in the UK, where about 60% of nurses were not satisfied with their work environment [93]. In Israel, nurses reported fear of treating infected patients, while remaining committed to their work [94]. A study published in 2018 by Satuf, Monteiro, Pereira et al. [95] evaluates the beneficial influence of job satisfaction on well-being and health. In a pandemic, this dependency is strained.

In a recent study, Möhring, Naumann, Reifenscheid et al. [88] showed that not enough research emphasis has been put on the severe social consequences brought by the pandemic. Neither in terms of how individuals behave among each other nor what is socially acceptable or preferred by society. This is particularly important concerning the stipulations on adhering to physical distancing and hygiene rules, i.e., minimizing personal contact to stop the virus from spreading [96]. To promote organizational health and well-being, a model for employee engagement focusing on conciliation, cultivation, confidence, compensation, and communication was proposed [97], combining many of the aspects other researchers have looked at in their work. Linking to this previous research, one of the key research contributions of this study is to provide new healthcare-sector-specific knowledge, expanding the existing research.

### 1.10. A New and Transtheoretical Model for CSR in the Healthcare Sector

Previously, rather little theoretical work was done in the area of CSR (e.g., [98]) with regard to experiences and concrete behaviors and there are a few different models being applied to social accounting [99]. Therefore, a well-developed model of health behavior change is applied to CSR, which will be described in the following and used for the empirical work of this study (e.g., [97]). 

While in general, CSR and sustainability research do not offer any standardized set of questions for the necessary compilation of data, this research field also still lacks appropriate models as a theoretical basis for such studies. Hence, the Compensatory Carry-over Action Model (CCAM) by Lippke [100,101] was chosen as a theoretical basis to go beyond cognitions with the expectancy violation theory [98] and rather understand behavior related to concrete actions. Moreover, few theories exist on holistic approaches combining different actions or behaviors that address energy consumption and waste management as well as physical distancing and hygiene rules. However, only if such a holistic understanding is achieved modeling also changes over time, a successful improvement strategy can be initiated or maintained. At the same time, the CCAM model is versatile and can be easily adapted to CSR and sustainability practices (see Figure 2).

The CCAM is among the first models enabling the comparison of behavior patterns. The underlying assumption is that the behaviors will help to reach defined goals, and individual behavior patterns which interact with each other. Those patterns leading to goal attainment appear more frequently than those hindering success [101]. Concerning the healthcare sector, the pandemic has had an impact on various sections as outlined in the following. Two CSR-related topics are high energy consumption and production of enormous amounts of waste. 

Looking at the CCAM, a hospital could decide on a higher level goal, such as being more sustainable in terms of reduced CO_2_ emission (i.e., energy consumption and waste outcome aka “carbon footprint emission”). The intentions would then be defined as reducing energy consumption and limiting the amount of waste by a given percentage or total number. The action plans put into place are, e.g., using more environmentally friendly products with a higher share of recycled materials and turning off machines and computers that are not in use. Both might be very challenging in daily routines but can also lead to financial gains and positive recognition for the efforts by both the employees and society. This transfer effect might create even more opportunities for sustainable behavior.

### 1.11. Aims of the Empirical Study

While the initial study’s focus was on one exemplified country, namely Germany, and its healthcare providers’ CSR activities, the pandemic’s influence on these activities is far reaching across sectors and country boundaries. Therefore, the research needs to be enriched by questions specific to places from all over the world, identifying how the pandemic was handled to better understand how crisis management works and what lessons can be learned for the future. A previous study in Pakistan found that the CSR engagement of a hospital, including the employees, is imperative for promoting workplace innovation and psychological safety [52]. Accordingly, the current study aims to provide in-depth insight into the effects, changes, and challenges of the COVID-19 pandemic and the motivation, satisfaction, and well-being of the employees. 

As shown above, the research on CSR during crises is very limited, the COVID-19 pandemic provides a crucial opportunity to explore this field of research. This study, therefore, contributes to filling the existing research gap, while at the same time giving specific insights into one sector by answering the *research question*: What CSR practices and activities have healthcare providers implemented in Germany and what effect does the pandemic have? 

This article makes two key contributions. First, it helps to gain a better theoretical and practical understanding of CSR activities in the face of a crisis. Second, the empirical research will expand the knowledge of the distinct fields of single actions within CSR and the crisis due to the pandemic. While we know already that there are interrelations, this study will contribute to a *holistic* understanding and guide for more effective encounters in the future. The following is structured as follows: the next section describes the methodology of the empirical study. The subsequent section reports the results obtained and analyzed, along with the synthesized categories. The last section closes with conclusions and future lines of research and practice.

## 2. Materials and Methods

To answer the research question, expert interviews were conducted to collect data during the second wave of the COVID-19 pandemic. We intentionally chose not to start during the first wave of the pandemic as more people obtained experiences over time and became aware that the actions required would not just be on a short-term basis but be sustainably relevant. However, the COVID-19 pandemic turned out to be an important influence on CSR activities, as well as on challenges and opportunities for the healthcare sector, and their employees’ motivation and satisfaction. The interviews were therefore amended by related questions following the method of participatory health research [102,103]. Concretely, data for the analysis was collected via expert interviews. To analyze the data, the software MAXDAQ was used as support to efficiently and correctly cluster all information according to topics. The number of concrete topics asked to report on in the interview and the volume of detail within the answers was high. To handle the input, using software for support proved to be feasible and practical.

Although the software was used, the method of data analysis is based on the so-called “qualitative text analysis process” by Kuckartz [104]. In phase one, all collected data is screened for important information that is worth looking into further. All expert interviews were carefully analyzed including checking the answers to each question for concise input. Though the software was used, the authors aimed to gain a first impression of the collected data by going through it by hand. The second phase focused on the development of the main topics. In this research, those topics were already pre-structured by the interview questions. Identified answers were consequently clustered in the “first coding process” (which is step three in the analysis). The following phases “compiling all information for each main category”, “creating sub-categories”, and the “second coding process” [105] were done with the help of MAXDAQ. Overall, two authors went through this process. Whenever the authors had different opinions about how to code a certain piece of information, all aspects were discussed, until a consensus was reached. This happened only very rarely.

In 2019, Germany counted 1.914 general care, psychiatric, or rehabilitation facilities [105]. The respective organizations are either public, private, or non-profit. The distribution varies depending on the different German states [106]. In addition, there are university hospitals that are usually part of the communal structure but are more independent due to the strong research commissions. Finding participants for a qualitative study during a pandemic was highly challenging. The research focused on finding participating facilities within all three care foci (general, psychiatric, or rehabilitation) and across the federal territory. Finding a provider that solely treats psychiatric diseases was not possible, but some of the chosen corporations offer psychiatric treatments, general care, and medical rehabilitation options. 

Overall, *N* = 18 German healthcare providers were recruited to participate in the study, based on these selection criteria, many of them operate various facilities. The first contact was made by phone. Most people either decided on the spot to participate or decline and only two asked for further information to forward it to the primary contact for the topic. Nine providers decided not to participate, usually due to time constraints. The other nine providers agreed to participate in the interviews, their characteristics can be found in Table 1. The providers’ parameters range from large corporate structures with hundreds of healthcare facilities to smaller providers with only a handful of houses to individual houses, divided among non-profit organizations and highly value-driven corporations. The data was collected between the middle of January and the middle of March 2021, during the high time of the second wave of COVID-19 infections in Germany. The length of the interviews varied from 35:44 min to 69:25 min. The interviewees’ companies and facilities have the following characteristics (compare Table 2):

The headcount ranges from 900 to 73,000 employees among the nine participating companies. The smallest participating healthcare provider has a yearly revenue of EUR 110 million and the largest company’s yearly revenue accounts for EUR 6 billion. Eight interviewees reported that they are part of a corporate group and only one hospital does not belong to a group. The results show that four study participants work for a profit organization, while five said they work for a non-profit organization.

## 3. Results

### 3.1. Coding of the Interviews

For the coding of the interviews, 27 codes were generated based on the interview questions. These codes yielded 651 coded segments. For this particular manuscript, the focus is on COVID-19-related results. These account for 211 coded segments, spread across the individual codes as shown in Table 3.

While the initial focus of this research was on CSR activities, COVID-19 and the second wave became the amended focus. Therefore, the focus was shifted to the COVID-19-related stresses.

### 3.2. Effects of the Pandemic

#### 3.2.1. Effects on CSR-Related Activities

Resources had to be allocated towards tackling the challenges of the pandemic whilst certain activities were put on hold, e.g., reducing waste, and using less plastic, based on legal and hygienic requirements. The amount of protective gear immensely increased, i.e., gloves and capes, most of it produced in China and Malaysia and transported to Germany. In Germany, this is hazardous waste and new ways of reducing and disposing of this type of waste need to be found. Simultaneously acquiring products with a smaller carbon footprint must be pushed. People stopped objecting to extensive use of energy when it comes to saving lives through energy-intensive high-tech medicine.

Those participants monitoring the entire climate balance or just the energy consumption will most likely be able to spot great differences in the results pre-, during, and post-COVID-19. With a shifting focus—away from reducing waste or energy consumption—the previously carefully monitored products, goods, and elements were suddenly so important for saving lives that all costs (even a negative impact on the climate) were justifiable.

The switch to online meetings had a positive effect on transparency and the feeling of being part of what happens. It is especially important for CSR activities, as they are more effective if accepted and supported by a larger number of employees. One interviewee reported that employees felt more included in the overall organization because they could participate in online events with topics beyond their everyday main duties. In addition, online events and meetings have a beneficial influence on the environment, because people do not need to commute, and emissions are reduced. In some cases, not having to travel for workshops and training meant a smaller time commitment and made participating easier. Nonetheless, in-person contact is sometimes essential, such as in fundraising, where the contact-reducing measures make reaching potential donors personally almost impossible and making the jobs of fundraisers more difficult. Not being allowed to have regular team meetings in person or updating the incoming shift caused challenges in the organization of the processes. Generally, the pandemic comes with many challenges for the ecological aspect of CSR especially.

Some, however, considered this pause to be a welcome break. It allows for structures and processes to be reviewed and developed. This is especially beneficial for CSR/sustainability departments that were in the stage of being formed or working on large sustainability projects. Many such activities used to be limited to central departments. Individual facilities appeared too consumed by caring for COVID patients and protecting patients and employees from the virus, there was no room for strategically pushing CSR activities.

At the same time, many measures connected to digitalization (communication, online events, video content) were in planning but not a top priority. In these cases, the pandemic functioned as a welcome booster for such topics. One interviewee reported that the general CSR work continued and so did the monitoring of the activities, although with fewer audits. 

#### 3.2.2. Effects of the Pandemic on Medical Care Structures

Regulations in a pandemic constantly change, therefore, employees have to be kept informed about things such as disinfection protocols, use of protective gear, test requirements, etc. Overall, working with safety equipment, the regulations tend to constantly change gear, and the awareness that becoming infected by the virus oneself is possible at all times increased the stress level among employees. The need for higher frequencies of communication and building task forces suddenly became apparent. Regulations changed so frequently that keeping up with visitation policies, childcare, professional training in person, group sizes for meetings, etc. proved to be a challenge. Adapting the operational processes was required.

Elective treatments and surgeries had to be canceled and postponed, leading to less work and more quiet stations in some hospitals. Patients were worried about going to a healthcare facility and especially to a hospital due to the fear of contracting the virus there. This behavior resulted in necessary treatments for heart attacks, strokes, appendicitis, and cancer—to name just a few—being postponed, leading to worsening conditions, “some will not recover (fully)”. Therefore, communicating that going to a hospital is safe, will be one of the main challenges after the pandemic (as an example for a concrete initiative please see Clinotel Patientensicherheit: www.patientensicherheit.clinotel.de, accessed on 18 July 2022).

At the same time, the pandemic forced healthcare providers to look for new ways of caring for patients. One concept that has already been fruitful is offering patients a choice of hospital and attending physician, even if that means that one physician needs to travel flexibly between facilities. During the pandemic, facilities that had COVID-19 outbreaks could be avoided by patients who urgently needed treatment as they were offered a bed in a different facility. Strengthening this development and at the same time decreasing the hurdles of transferring patients between facilities and institutions became imperative. As for events and regular offers for employees, those had to be either canceled altogether or switched to online formats. As contact was often not permitted, the variety of options was reduced, but classes for resilience and online sports courses still took place.

Quickly going in new ways can be beneficial, being most prominent when looking at digitalization—from different (new) forms of communication to online training and workshops—having meetings and events online created new opportunities. Things that used to take a very long time are speeding up due to the pandemic; for example, home office or working digitally with video conferences. Although this meant and still means that personal contact was/ is reduced to a bare minimum. One recognizes “Hey, we can do that, it works.” Hence, once the pandemic is overcome, “certain things will not be turned back”, e.g., digital meetings with family and accompanying persons for individuals in healthcare facilities, as they allow for more frequent contact.

The work has the same high quality even with changes in structures, habits, and processes, and the agility and engagement exponentially increased. Nonetheless, some processes were put on hold, especially in the field of CSR and sustainability. For example, building new management structures or creating and publishing CSR reports, because resources had to be allocated towards dealing with the ramifications of the pandemic. Apart from that, priorities changed, which often affected CSR-related activities. Decreasing the amount of waste produced in a hospital is probably the most obvious. Due to enormous amounts of protective gear and requirements to use disposable crockery, the waste increased drastically, and actions were required.

Problem-solving was one of the greatest tasks, childcare had to be organized, personnel had to be shifted between stations, and sometimes even countries, as some nurses and doctors from one provider helped in Spain during the first wave. The interviewee articulated: “Employees were desperate because they had such a patient overload that they did not know how to take care of them anymore. Situations were even worse than what the media portrayed. Sending personnel from Germany was one way to offer support.” Thus, the pandemic can overall be considered a driver for sustainable changes and innovation.

### 3.3. Change in Times of Crisis

#### 3.3.1. Changes in the Well-Being, Satisfaction, and Motivation of Employees during the Pandemic

Working with patients with severe cases of COVID and seeing the virus’ devastating effects, made the danger more real for employees, increasing the stress. The insecurity due to rapidly changing processes also decreased well-being. Some reported that the well-being, satisfaction, and motivation differed among employees from various departments. In departments having had very few patients since the beginning of the pandemic, the emptiness was perceived as a burden, too. While at the same time, others, especially in the intensive care units and on the COVID-19-stations, were highly overworked. Everyone, in their way, was exhausted and friable, by “we need to continue”, the “pandemic is not over yet”, causing many to be on the edge. Others see the same patterns within their employees as in the general public, differing based on individual characteristics.

To increase the mood and motivation, and support the employees, some arranged in-house events such as star cooks providing special meals or offering training online to get new ideas. “With regard to training employees, we have taken major steps forward”, one interviewee said, explaining that online options are less time-consuming and accessible for more employees. The work ethos is high, particularly because society experienced how important the healthcare sector is and the encouragement for employees in this sector was immense at the beginning. However, a small COVID-19 bonus does not compensate for the work the employees have been doing for months and the fear they lived in. With people becoming less careful, some healthcare workers felt that people were having fun at their expense.

Two experts suggested that the impact was not substantial because employees were so experienced in what they were doing. They moved even closer together, creating a stronger bond of belonging. One said that the satisfaction among administrative staff increased because they were able to work from home, allowing them to help their children with homeschooling.

Employee surveys were conducted to evaluate how the employees came through the first nine months of the pandemic, what they need to get through the next, what they wish for in the future, what challenges they faced, and the positive aspects the pandemic offered. The results have yet to be analyzed. Many suggest that the well-being and satisfaction of employees will increase once they are vaccinated since they will have more protection and are less likely to fall seriously ill, which both reduce psychological distress.

#### 3.3.2. Changes in the Internal and External Communication Due to the Pandemic

Regarding internal and external communication, the interviewees reported the following changes (Table 4):

In summary, digitalization played an important role in communication. Internally, the information exchange, meetings, and briefings happened digitally. External communication was influenced by the presence of digital media that produced loads of information at a rapid speed. Although transparency increased due to new channels of communication and higher frequencies, giving all relevant information without creating fear was perceived to be a great challenge.

#### 3.3.3. Structural Changes Due to the Pandemic

Several interview partners reported that the pandemic brought about quite a few structural changes. The new and different forms of communication introduced by the pandemic changed the ways employees are informed about changes as communication became much broader and dissolved bottle-neck structures. High hopes are put on this alteration being sustainable and incorporated into people’s minds. Many things had to be changed ad hoc, without lots of preparation time, which is very different from the otherwise lengthy change processes in the healthcare sector. 

Decision-making processes also changed and increased in speed. Different situations in various facilities and regions required the possibility to react individually and arrangements were made via video conferences without being present in person. The entire process by which patients are channeled through a facility had to be restructured to include tests and to separate infected patients from uninfected patients upon entry into the facility. The fact that patients come and leave alone (without a relative being present) also posed a greater need for support from the personnel. 

The focus on specializations became very important and the pandemic is a driver of the change, pushing this phenomenon. Less contact forced the healthcare sector to structure a new channel for revenue: digital healthcare and telehealth. New means of treatment and patient communication started to develop with the need for novel out-patient treatment. However, insurance companies only covered digital doctor-patient sessions for a certain amount of time. The healthcare sector still has a long way to go in terms of establishing well-rooted care options, especially in Germany.

One interviewee said while digitalization increased in speed, which affected organizational and treatment structures, all in all, the pandemic did not force the healthcare provider to introduce anything drastic. It was not the first pandemic they were handling. Ebola, swine fever, and the EHEC-crises prepared them, and emergency plans were in place to react swiftly and deliberately by redistributing resources and work. Another one also said that no structural changes were visible.

#### 3.3.4. Opportunities and Challenges in Everyday Routines

Although the pandemic caused a lot of challenges, the number of opportunities it offered was probably equally as high. The interviewees reported the following challenges and opportunities for their facilities (Table 5).

“Generally, all formats of exchange became digital”, prompting employees to use different and new ways of communication, resulting in all personal contacts being strictly reduced. Creating a familiar sense of reachability and proximity was a challenge in the beginning, but the changes were quickly accepted. For some, this was and still is immensely demanding, because regulations often changed and, for example, hygienic requirements have become stricter. Daily routines changed drastically for some, causing them to show extremely high flexibility in terms of working hours and workplaces—switching between stations, and helping out where the manpower was needed. Nonetheless, “everyone pushes through and keeps going”.

The reported challenges and opportunities demonstrate that crises in organizations have negative and positive consequences and provide the opportunity for improving the technological infrastructure and the organizations themselves. Psychological safety of individuals and the overall health and well-being of groups and organizations were affected, too, and require appropriate attention [82].

All in all, the need for rapid changes and agile processes posed challenges and opportunities at the same time. The healthcare sector showed its versatility in managing obstacles: (1) keeping up with regulations; (2) guaranteeing enough personnel was available at all times to ensure the patients’ care; (3) taking care of employees’ needs during these strenuous times; (4) keeping the public informed; (5) and reducing CSR affords. 

Reported opportunities were: (1) digital communication, which created greater transparency and allowed for higher participation; (2) the focus on core competencies to provide the best possible care and fostering interdisciplinary teamwork to solve problems; (3) the reduction in the carbon footprint due to fewer travels; (4) putting big centers for care and research into a different spotlight highlighting their benefits; (5) and the realization that the importance of sustainability steadily increases.

Although many CSR activities had to be put on hold, waste production spiked, and energy consumption increased rapidly, the value of such activities has not been forgotten. On the contrary, many have reported that emphasizing sustainability in all situations will have to play an important part in the future—with regard to saving the environment, being economically efficient, being socially responsible, and being an attractive employer. Taken together, the findings can be aggregated as described in Figure 3: 

These major findings are crisis-related, which means they can be adapted to other crises similar to the COVID-19 pandemic. Therefore, the results generate valuable input for managing future challenges through adapted CSR including cooperative leadership, crisis management, strategic management, and crisis communication.

## 4. Discussion

This study looked into the challenges and opportunities the German healthcare sector faced during the second wave of the COVID-19 pandemic, with a special focus on corporate social responsibility (CSR) activities, based on the Compensatory Carry-over Action Model (CCAM). The results offer insights into changes, challenges, and opportunities care providers are confronted with. Connected are motivation, well-being and satisfaction of employees that are essential elements of an organization’s attractiveness, e.g., [18,27]. The different approaches to handling them show the opportunities to learn from the pandemic, change behavioral patterns, and open a discourse about the way forward. 

Especially through the lens of CSR, the study shows how gravely the pandemic impacted different CSR activity efforts throughout the entire sector but also beyond CSR the work in general in face of crisis, and options of CSR to help to cope with crises. To continue the path towards a more sustainable healthcare sector, CSR activities are either even higher on the priority list or the first projects to be stopped or abandoned during a crisis. This results in the previous actions and behaviors being either turned back or stopped. CSR drifted out of corporate and public focus, and organizations fell behind on their commitments. However, at the same time, CSR was found to facilitate sustainability of quick coping actions. Given that various theories and models exist for the concept of CSR, the application of each model might yield slightly different results. 

A noteworthy finding is that effective communication is a key element in disseminating important information and creating stability amongst the workforce, an aspect that affects management behavior, CSR activities, and crisis management [67,68,69,74,97]. Well-being, satisfaction, and motivation were perceived better when communication was efficient. While many activities were canceled due to social distancing regulations, online events helped to fill the void. Although not being able to meet in person poses a tremendous challenge, switching to online formats offered new opportunities for knowledge transfer and a new way of thinking. The results tie in with the findings of previous research (e.g., [16]) showing that the right form of communication is essential to the well-being of employees working from home, given that it decreases the stress brought on by technology. At the same time, information about safety regulations and protection measures did not seem to reduce anxiety, leading to essential treatments being postponed and putting psychological safety in question [82]. It is important to note here that psychological safety was demonstrated to be imperative for mediating the effect of perceived organizational support on work engagement (perceived organizational support and work engagement). Concerning the introduced CCAM model, communication can function as a positive transfer of intentions. However, the behavior of those being informed can pose a challenge if the efforts do not yield the desired outcome.

Overall, demands appear to have increased while resources were challenged. CSR-related behavior and the respective activities were intended to be pushed, but many interviewees reported that several activities could not be pursued to the planned extent. Nonetheless, the realization of the importance of continuously pursuing these activities is immensely valuable for the strategic orientation of healthcare providers—once the pandemic gives room to look at the future [66,84,85,86,87]. 

Others argue that climate change, energy consumption, and waste management have an impact on health and should be the focus of companies and hospitals. Transforming this urgency into actively tackling these issues, with the help of adequate monitoring instruments, becomes the challenge of the post-COVID times [72,73]. The first ideas to tackle the above have already been implemented. They can, at any time, be refocused and transformed to create a more lasting impact. Nevertheless, sector-wide, standardized measures have to be established to generate a sustainable, long-term impact [8,16,17]. 

The CCAM can function as a basis for outlining the different behaviors in question, their concrete steps for becoming realized, and how they can work together in concert. For instance, general CSR activities as one stream of action and digitalization as another should be pursued concurrently: Firstly, intentions towards both have to be formed and then planned in detail to actively change individual, group, and organizational behavior. Transfer between the two streams or behavioral domains could be facilitated using synchronized communication and the development of hope, expressed as common ground in the interviews. 

The finding that crisis generates momentum can be seen as experiences facilitating behavioral change and overcoming (previous) resistance. Looking at the CCAM, a hospital could decide on a higher level goal, such as being more sustainable in terms of reduced CO_2_ emissions (i.e., energy consumption and waste outcome). We have seen that corresponding coping mechanisms were acquired or fostered. 

Digitalization, especially in crisis communication, matching other studies, and when exchanging safety information, was reportedly the factor getting the most support through the pandemic, creating greater transparency. This can function as a transfer of cognition elements, leading to the exchange and facilitation of experiences. Although the initial plan to pursue CSR activities could not be fulfilled, the new experiences nonetheless had a positive impact. 

Using technology and being aware of its positive effects increase the opportunities for reaching goals [67]. Through new regulations, the German government [107,108] has paved the way for faster digitalization in the healthcare sector. The necessity for digital integration, communication, and knowledge transfer has become evident during the pandemic and was strongly fostered [109]. The created momentum can help to push the provision of medical services and the way medicine is practiced in new spheres. It remains to be seen how sustainable the changes in digitalization are, given that the presence of this specific pandemic was the driving factor.

Although the pandemic was, and still is, an extreme situation, with streamlined offers, coping with uncertainty and heightened workload became manageable as also revealed in other studies [67,69,97]. Overall, being in the spotlight is considered to be positive because it allowed healthcare providers to show competencies that will have a beneficial impact on their image [11,12]. In some cases, the desire to help people in need resulted in higher application numbers for training to be a healthcare professional or assistant. This might be a valuable asset in tackling the issue of not finding enough personnel.

Thus, one can argue that the pandemic functioned as a driver for change in some cases while hindering progress in other areas. External influences such as a pandemic can cause great challenges leading to not being able to fulfill a plan and reach a goal. Some valuable insights carry over, past experiences lead to new ways, and the transfer of information (i.e., new communication channels), knowledge (i.e., digitalization), and ideas (i.e., creating specialty areas) is a step forward.

Besides the results concerning CSR-related strategies which support conclusions of other researchers (e.g., [2,3,8,13,14,15,16,17]), the findings show the deep impact of the pandemic on the health and well-being of people working in the healthcare sector: This study’s results confirm these findings to an extent: (1) the immense stress while working in constant danger of contracting a virus, having changing workloads, (2) dealing with rapid changes in regulations, (3) and living with a complete disruption of everyday life can influence the satisfaction, well-being, and motivation negatively. This validates findings in other studies also beyond CSR such as the one by Schmiedhofer et al. with obstetric healthcare workers assisting births under the condition of the COVID-19 pandemic in Germany [110]. 

To counteract the negative forces, healthcare providers have looked for ways to empower their employees and support their well-being—a trend that can also be witnessed in other countries, e.g., [51,52]. The necessity for this behavior was also proposed by various other findings [67,97]. Offering special meals or specialized training, providing child-care options, and keeping employees up to date regarding the newest political, medical, and societal developments were all perceived to have a positive influence on the employees.

The study’s main limitation is the number of participants which seems to be rather small at the first glance. However, given the fact that so many interviewees are working for a corporate group, they provided a vast array of knowledge. The interviewees were able to give insights into various facilities and with the fact that facilities of different sizes, as well as profit and non-profit organizations participated, the results are nonetheless representative. In addition, the results are similar to those generated in other countries [26,27,28,29,30,31,32,33,52,59,60,91,92,93,94,95]. Chances are high that some aspects (changes in communication and digitalization) can be generalized across sectors and even countries, however, this hypothesis would need to be verified in future research.

As for practical implications, the study shows that most healthcare providers have similar experiences during the pandemic, no matter what size or field of specialty. Furthermore, the strain that the pandemic put on healthcare workers is immense, impacting their quality of life and well-being, satisfaction, and motivation. This knowledge can help support employees, make them more resilient, and provide options for finding a balance again to secure a strong and content workforce. Crisis management and crisis communication should take the larger picture into account, including coordinated strategic management.

The healthcare sector was put into a new spotlight due to the pandemic. This spotlight can be harnessed as a driving factor to foster sustainable changes and developments. Conducting further research on this topic can provide even more insights into the effect that the crisis had on healthcare providers. These learned lessons can help to manage other crises, too, by means of transferring knowledge and developments, evidence- and theory-driven.

## 5. Conclusions

The pandemic altered the way business is conducted, and especially the healthcare sector was, and still is, under immense pressure. The allocation of resources–equipment, finances, and personnel—proved to be a huge task. Those who have already had monitoring instruments in place will be able to analyze in-depth, how drastic the changes were and are pre-, during, and post-COVID-19. The interconnectedness to provide the best possible care grew because not every facility was able to provide any treatment. Focusing on specific competencies created hubs of specialization. In general, solving any arising problem became the top priority. Depending on the number of cases and hospitalized patients in need of extensive care, the stress for healthcare personnel increased dramatically, leading to lower job satisfaction and decreased overall well-being. As the compensation for the hard work was minimal, if at all, also the motivation of healthcare staff decreased over time.

Due to the pandemic, many processes and methods were reevaluated, causing structural changes, as minimizing personal contact became one of the highest goals. Hence, decision structures were completely altered due to remote work and virtual instead of in-person meetings. Communication barriers decreased, opening communication channels for all employees by means of managerial communication and effective leadership [69,70]. The degree of digitalization grew exponentially, creating new work formats. Online events facilitated broader participation across locations and departments. Switching to online communication gave greater access to information and events because employees could participate without needing to travel but at the same time, this might cause challenges as demonstrated before [69,70]. Reduced travel costs and more efficient use of time were two of the positive outcomes.

As for CSR activities, the pandemic was a blessing and a curse at the same time. While the amount of waste and single-use plastics increased dramatically, online events and digitalization fostered sustainable development. This can be understood and facilitated by making use of theories such as the CCAM which was used as theoretical background for this study. The article’s contributions to the CSR literature are the use of an innovative theory and its empirical exploration during times of crisis, i.e., the COVID-19 pandemic on CSR activities of healthcare providers with a special contribution to the monitoring of these activities.

One major finding is the fact that many CSR activities were temporarily put on hold during the pandemic, showing that they are not yet fully incorporated into everyday work but rather considered as add-ons or nice-to-have activities. Especially environmental CSR activities were reduced or completely stopped, much to the dismay of employees. Waste management and energy usage are the most prominently affected areas. 

To fully integrate these environmental aspects into the business structures, CSR activities need to become an essential element of the business strategy that will not be disregarded once things get tough or crises develop. The findings show that CSR might be a highly discussed topic within organizations but is not as firmly anchored within core business practices as it often seems. Hence, this research adds to the understanding of CSR practices in organizations and contributes to filling the research gap on how CSR is handled in the business context. The exact effects of the pandemic must be determined in the course of the coming months and years.

## Figures and Tables

**Figure 1 ijerph-20-00368-f001:**
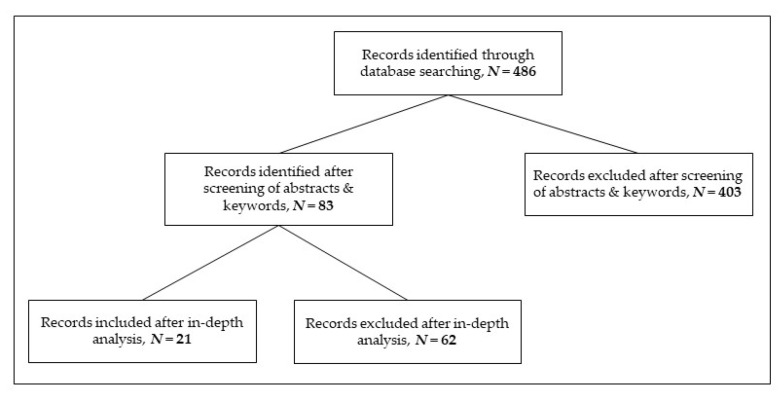
Inclusion and exclusion of manuscripts.

**Figure 2 ijerph-20-00368-f002:**
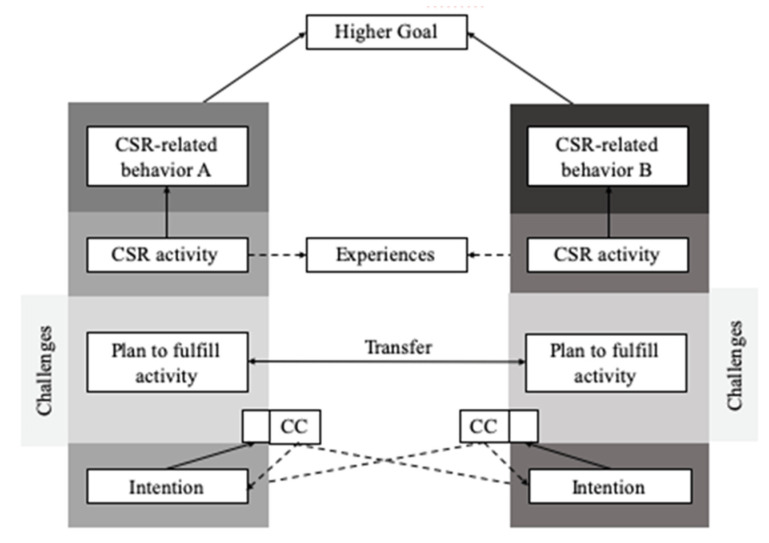
The Compensatory Carry-over Action Model (CCAM) applied to CSR (on basis of [101]); CSR = corporate social responsibility; CC = compensatory cognitions; behavior A can be, for instance, energy consumption and waste management; behavior B can be, for example, physical distancing and hygiene rules to stop pathogens to spread.

**Figure 3 ijerph-20-00368-f003:**
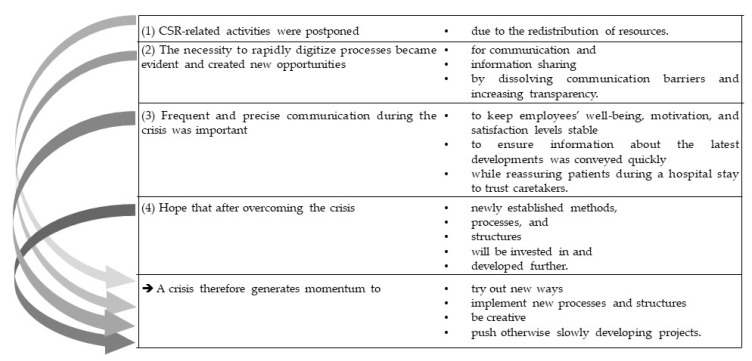
Aggregated research findings. Content was revealed by means of a qualitative text analysis process [104] employing MAXQDA.

**Table 1 ijerph-20-00368-t001:** Overview of monitoring insights from the systematic review.

General Understanding	Convergence of Standards	Monitoring Institutions	Monitoring Mechanisms	Acknowledging Influence	Monitoring Costs	Monitoring of Managers
- monitoring economic impact of projects defined in corporate philanthropic strategy (315) - monitor social impact of projects defined in corporate philanthropic strategy (315) - CSR activities have to satisfy stakeholders (2054) - CSR can only be effective if all stakeholders are involved in the implementation (155)	- monitoring to enhance the balance sheet (327) - evolving sustainability reporting (225) - existing standards do not include monitoring aspects (234) - existing standards do not include sanctions (234) - using elements of agency theory (20) - applying aspects of resource dependence theory (20) - CSR is seen as a voluntary standard (25) - CSR as an important driver for SDGs (25)	- free-rider issues in larger boards (49) - higher coordination costs in larger boards (49) - directors need strong incentives (49) - directors need experience (49) - security analysts function as third-party monitoring bodies (260) - establishing CSR committee for monitoring purposes (44) - strong CSR commitment associated with good governance (44) - financial analysts function as third-party monitoring bodies (21) - more analysts = more monitoring = more information distributed (21) - monitoring often outsourced (234) - monitoring only important for those with large shares (52)	- measuring the impact of defined goals (327) - no clear instrument available to prevent over-investing in CSR (260) - CSR committees in themselves are an important mechanism (20) - regional characteristics functioning as a monitoring mechanism (20)	- board members need to acknowledge their influence (44) - reporting and accounting systems are influential tools in monitoring (234) - all corporate-related decision-makers (e.g., board members, shareholders, analysts, regulators, etc.) have influence (260) - companies gave their own influence over their external interests (155)	- dividing the work and outsourcing some, is efficient cost management (234) - monitoring is costly, due to the needed depth of analysis (52)	- scandals clearly show monitoring deficits (41) - good governance shows managers’ behavior (14) - good governance makes the investment into CSR visible (14) - information about CSR costs levels managers’ personal opinions (144) - CSR reports can identify managers’ wrongful actions (144) - good monitoring reduced the risk of value destruction (144) - awareness about the importance of CSR reports can influence decisions (144)

**Table 2 ijerph-20-00368-t002:** Characteristics of the corporations participating in the study.

Participant	Headcount	Yearly Revenue (EUR )	Part of a Corporate Group	Non-Profit or Profit Organization
1	13,500	1.2 billion	No	Profit
2	16,000	872 million	Yes	Non-profit
3	73,000	6 billion	Yes	Profit
4	30,000	1.632 billion	Yes	Non-profit
5	2600	183 million	Yes	Non-profit
6	1800	110 million	Yes	Profit
7	3200	200 million	Yes	Non-profit
8	900	950 million (group)	Yes	Non-profit
9	4000	400 million	Yes	Profit

**Table 3 ijerph-20-00368-t003:** Codes and the respective number of related segments.

Code	Number of Segments ^1^
COVID-19—Effects of the pandemic	29
COVID-19—Effects on CSR-related activities	27
COVID-19—Challenges the pandemic brought along	28
COVID-19—Changes in the internal and external communication by the pandemic	27
COVID-19—Where the pandemic forced structural changes	17
COVID-19—Changes in everyday routines	7
COVID-19—Opportunities the pandemic offered	36
COVID-19—Changes in the well-being, satisfaction, and motivation of employees during the pandemic	31

^1^ Segments were determined by means of qualitative text analysis process [104] employing MAXQDA.

**Table 4 ijerph-20-00368-t004:** Communication changes ^1^—internal and external.

Internal Communication	External Communication
Higher frequency	Crisis communication took up more time and resources than usual
Greater importance	Frequency increased only in some cases
More digitalized (e.g., meetings, information exchange, briefings)	Press representatives and relatives demanded updates more often → exponential increase
Greater transparency through digitalization and higher frequency	Media hype around information caused a shorter half-life period of information
Introduction of task forces	Communicating to reduce insecurities among the population and potential patients remains a huge task
Difficulty in giving out the necessary information, because official bodies took so long to pass regulations	Discrepancy between being transparent without reinforcing fear was very strenuous

^1^ Changes were determined by means of qualitative text analysis process [104] using MAXQDA.

**Table 5 ijerph-20-00368-t005:** Reported challenges and opportunities ^1^ of the pandemic.

Challenges of the Pandemic	Opportunities of the Pandemic
A shift in priorities: overcoming the pandemic became THE goal	More people can participate in online meetings and events
CSR activities were either canceled, paused, or drastically reduced	Online meetings and events reduce carbon footprint (fewer travels) and reduced time commitment
Informing employees that CSR activities had to be reduced due to the pandemic was very difficult	Higher frequency of communication increases transparency and includes more people
Quickly acquiring enough vaccines for the employees	Agility was increased in every aspect
The vaccine was seen as a relief, waiting for it was another stressor	Digital communication with increased frequency led to a stronger feeling of closeness and togetherness
The fluctuation between workload in different departments was immense	Working together with different disciplines was increased
When personnel was scarce, caring for patients became difficult	Working from home on a large scale was perceived as innovative for the sector and as very positive
Attracting enough personnel is generally difficult in the German healthcare sector, the pandemic put an additional strain on these efforts	Digital communication encouraged more employees to participate and offer ideas and opinions
Different regulations between federal states and rapid changes in regulation required agility and constant communication	In person contact was more cherished
Ensuring patients felt safe to come in for treatment, making sure they did not contract the virus	Big hospitals are the center of attention in a pandemic, which might be a chance to change the image and generate greater recognition
Patients fear of coming in for treatment worsened conditions or closed windows of opportunity for treatment	Pandemic pushed things that would have otherwise not been pursued (e.g., rapid digitalization)
Focusing on centers of specialization	Increase in outpatient treatment, reducing hospital stays
Finding new growth potentials	Digitalizing and condensing the healthcare sector creates new business opportunities
Feeling that the German government uses the pandemic to cleanse the healthcare sector causes the quality of care, staff, and patients to suffer	The necessity for using resources sustainably (materials, energy, personnel) became apparent

^1^ Challenges and opportunities of the pandemic were determined by means of qualitative text analysis process [104] utilizing MAXQDA.

## Data Availability

The original data are not available for data protection reasons.

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
