# Peer review of "Impact of the COVID-19 Pandemic on CSR Activities of Healthcare Providers"

_ijerph, 2022, doi:10.3390/ijerph20010368_

Round 1

Reviewer 1 Report (Previous Reviewer 1)

NO comments 

Reviewer 2 Report (New Reviewer)

The authors of the article Impact of the Corona Pandemic on CSR activities of healthcare providers have addressed an important and momentous topic today. The article is worth publishing. It meets all the requirements. The structure of the text was preserved, and the appropriate division. I have no comments. 

This manuscript is a resubmission of an earlier submission. The following is a list of the peer review reports and author responses from that submission.

Round 1

Reviewer 1 Report

Article lacks the consistency, depth and flow. Title seems too long and unnecessary words to remove. Introduction and discussions are inconsistent. Manuscript needs major revisions. 

Author Response

AUTHORS: We are very grateful to all three reviewers for their helpful and encouraging comments. We considered the suggestions in detail and revised the manuscript accordingly. Please see attached the original paper with the revisions highlighted in track changes. In addition, in the following, we give quotes from the revised manuscript referring to lines in the manuscript without track changes.

Reviewer 1: Manuscript needs major revisions. Article lacks the consistency, depth and flow. Title seems too long and unnecessary words to remove. Introduction and discussions are inconsistent. Manuscript needs major revisions.

AUTHORS: We have tried our very best to improve the manuscript according to all reviewers’ comments and suggestions and very much hope we addressed all points sufficiently. We improved the consistency, depth, and flow. The title was shorted and now is wording “Impact of the Corona Pandemic on CSR activities of healthcare providers”. The introduction and discussions were changed to be consistent, please see the tracked changes for all edits.

Reviewer 2 Report

This is an interesting paper and the authors have collected data that has the potential to give us new insights.  I have a few suggestions that I think might help strengthen the paper:

1. There are some spots in the paper where the writing is difficult to follow and distracts from the content.  Two examples:  (1) The first sentence of the abstract is difficult to understand and should be revised.  (2) Later in the paper (Figure 2, and in lines 40, 474, and 506), the word "always" is used inappropriately and should be removed.  "Always" means that nothing else ever happens, and that is not what you have shown nor (I hope) what you mean to claim.  Removing the word "always" throughout the paper would address this issue.  I recommend also having someone go over the whole paper for clarity of language.  The more clear the language, all else equal, the more likely that the paper will be widely read and cited.

2. The relevance of the CCAM to the paper was unclear to me, as was its value in this context.  I do not see how this model relates to your study.  If it were removed from the paper, what would be lost?

3. You should review the CSR literature, since this article is about the effects of the pandemic, as a crisis, on CSR.  Given that both the journal and the article pay particular attention to environmental sustainability, you might want to look in particular at environmental CSR in your review.  But without knowing what the literature has already said about CSR in crises, it is difficult for you to show your paper's contribution.  What specific gap in the literature does this paper fill?

4. You interviewed many people.  Your conclusions get pretty broad, and I would like to see more of the detail that you found, since one of the benefits of qualitative research across a small number of businesses is precisely the richness of the details.  I recommend a few direct quotes, and maybe a few short stories about specific incidents.  That would add value.

5. The digitization part is interesting.  It is not directly related to CSR, but could use more explanation (for example, are digital sessions with patients still desired?), and I think you could tie it directly to CSR more (discuss the impact on patients and employees of more digital sessions, the impact on the environment, etc.).  You might also expand the rest, emphasizing that the digitization that happened this time was a result of the specific nature of this crisis.

6. You need to be careful to show the reader that with a sample of nine companies in Germany responding to one crisis, you don't think you've provided a definitive answer to the research question.  This relates in part to my comments about using the word "always," but if you think these findings generalize to other companies in other countries, other industries, and other crises, you need to call for future work to test whether this is so.  

7.  If I were to summarize the paper's findings, I would say that during the crisis, these companies put aside CSR efforts to focus on managing the crisis, and that some of the solutions they developed might continue to be useful after the crisis.  This seems reasonable, but not new.  I suggest that the revision should make more clear what the contribution is (c.f. my comment #3).

I hope these comments help.  Good luck!

Author Response

Reviewer 2: This is an interesting paper and the authors have collected data that has the potential to give us new insights.  I have a few suggestions that I think might help strengthen the paper:

AUTHORS: Thank you very much for your appreciation! We benefitted very much from your suggestions and believe the paper is now significantly strengthened.

Reviewer 2: 1. There are some spots in the paper where the writing is difficult to follow and distracts from the content.  Two examples:  (1) The first sentence of the abstract is difficult to understand and should be revised.  (2) Later in the paper (Figure 2, and in lines 40, 474, and 506), the word "always" is used inappropriately and should be removed.  "Always" means that nothing else ever happens, and that is not what you have shown nor (I hope) what you mean to claim.  Removing the word "always" throughout the paper would address this issue.  I recommend also having someone go over the whole paper for clarity of language.  The more clear the language, all else equal, the more likely that the paper will be widely read and cited.

AUTHORS: Thank you very much for this input. We have done our best to clarify the language and improve the overall reading flow addressing your points above and also in general. Regarding the clarity of language, we asked an additional native speaker and an expert in the field to proofread and accordingly we have improved the paper (see her mentioned in the acknowledgments: “Furthermore, the authors want to thank Ronja Bellinghausen, Grace Wire and Genevieve Barlow for the proofreading and editing of the manuscript.”, lines 650-652).

Reviewer 2: 2. The relevance of the CCAM to the paper was unclear to me, as was its value in this context.  I do not see how this model relates to your study.  If it were removed from the paper, what would be lost?

AUTHORS: We are very sorry that the main contributions of the paper were not communicated well, which we have adapted: One main contribution of the paper actually is the theoretical work employing the CCAM and the empirical work testing it. We extended the according sections to make this clearer:

„This paper aims to review the crises of the COVID-19 pandemic and apply a theoretical backdrop to highlight the connection and relationship to CSR.“ (lines 58-60)

„In addition, little theoretical work was done in the area of CSR (e.g., [49]), nonetheless different models are applied to social accounting [50]. However, as these models do rarely relate to experiences and concrete behaviors. Furthermore, they were found to be only theoretical developed which lacking in data. Therefore, a well-developed model from health behavior change is applied to CSR, which will be described in the following and used for the empirical work of this study (e.g., [42]).

While in general, CSR and sustainability research do not offer any standardized set of questions for the necessary compilation of data, this research field also still lacks appropriate models as a theoretical basis for such studies. Hence, the Compensatory Carry-over Action Model (CCAM) by Lippke [51-52] was chosen as a theoretical basis to go beyond cognitions with the expectancy violation theory [49] and rather under-stand behavior related to concrete actions. Moreover, few theories exist on holistic approaches combining different actions or behaviors that address energy consumption and waste management as well as physical distancing and hygiene rules. However, only if such a holistic understanding is achieved modelling also changes over time, a successful improvement strategy can be initiated or maintained. At the same time, the CCAM model is versatile and can be easily adapted to CSR and sustainability practices (see Figure 1). “ (lines 126ff).

„The CCAM is among the first models enabling the comparison of behavior pat-terns. The underlying assumption is that they will help to reach defined goals and individual behavior patterns which interact with each other. Those patterns leading to success appear more frequently than those hindering success [52]. With regard to the healthcare sector, the pandemic has had an impact on various sections as outlined in the following. Two CSR-related topics are high energy consumption and production of enormous amounts of waste. Looking at the CCAM, a hospital could decide on a higher level goal, such as being more sustainable in terms of reduced CO² emission (i.e. energy consumption and waste outcome aka carbon footprint emission). The intentions would then be defined as reducing energy consumption and limiting the amount of waste by a given percentage or total number. The action plans put into place are, e.g., using more environmentally friendly products with a higher share of recycled materials and turning off machines and computers that are not in use. Both might be very challenging in daily routines but can also lead to financial gains and positive recognition for the efforts by both the employees and society. This transfer effect might create even more opportunities for sustainable behavior.“ (lines 147ff)

„First, it helps to gain a better theoretical and practical understanding of CSR activities in the face of crisis. “ (lines 177-178)

„Concerning the introduced CCAM model, communication can function as a positive transfer of intentions. However, the behavior of those being informed can pose a challenge if the efforts do not yield the desired outcome.

Overall, demands appear to have increased while resources were challenged. CSR-related behavior and the respective activities were intended to be pushed, but many interviewees reported that several activities could not be pursued to the planned extent. Nonetheless, the realization of the importance of continuously pursuing these activities is immensely valuable for the strategic orientation of healthcare providers – once the pandemic gives room to look at the future. Others argue that climate change, energy consumption, and waste management have an impact on health and should be the focus of companies and hospitals. Transforming this urgency into actively tackling these issues becomes the challenge of the post-COVID times [47, 48]. The first ideas to tackle the above have already been implemented. They can, at any time, be refocused and transformed to create a more lasting impact. Nevertheless, sector-wide, standardized measures have to be established to generate a sustainable, long-term impact.

The CCAM can function as a basis for outlining the different behaviors in question, their concrete steps for becoming realized, and how they can work together in concert. For instance, general CSR activities as one stream of action and digitalization as another should be pursued concurrently: Firstly, intentions towards both have to be formed and then planned in detail to actually change individual, group, and organizational behavior. Transfer between the two streams or behavioral domains could be facilitated by means of synchronized communication and development of hope, ex-pressed as common ground in the interviews. The finding that crisis generates momentum can be seen as experiences facilitating behavioral change and overcoming (previous) resistance. Looking at the CCAM, a hospital could decide on a higher level goal, such as being more sustainable in terms of reduced CO² emissions (i.e. energy consumption and waste outcome). We have seen that corresponding coping mechanisms were acquired or fostered.“ (lines 505ff)

Reviewer 2: 3. You should review the CSR literature, since this article is about the effects of the pandemic, as a crisis, on CSR.  Given that both the journal and the article pay particular attention to environmental sustainability, you might want to look in particular at environmental CSR in your review.  But without knowing what the literature has already said about CSR in crises, it is difficult for you to show your paper's contribution.  What specific gap in the literature does this paper fill?

AUTHORS: Thank you for this tip. We have looked at the literature and included relevant research findings. Interestingly, most CSR research looks at all three major elements of the concepts, combining the social, environmental, and organizational realms. We hope that the context is now even clearer. Please see the multiple revisions in the manuscript highlighted.

Reviewer 2: 4. You interviewed many people.  Your conclusions get pretty broad, and I would like to see more of the detail that you found, since one of the benefits of qualitative research across a small number of businesses is precisely the richness of the details.  I recommend a few direct quotes, and maybe a few short stories about specific incidents.  That would add value.

AUTHORS: This is a noteworthy point, thank you for raising it. We have included a couple of direct quotes and a story. However, we refrained from adding too many direct quotes in order to keep the manuscript as concise as possible.

The sections now read:

  • “One recognizes “Hey, we can do that, it works.” hence, once the pandemic is overcome, “certain things will not be turned back”, e.g. digital meetings with family and friends for people in healthcare facilities, as they allow for more frequent contact.” (lines 328-331);
  • “The interviewee articulated: “The interviewee articulated: “Employees were desperate because they had such a patient overload that they did not know how to take care of them anymore. Situations were even worse than what the media portrayed. Sending personnel from Germany was one way to offer support.” Thus, the pandemic can overall be considered a driver for sustainable changes and innovation.” (lines 343-347);
  • “With regard to training employees, we have taken major steps forward”, one interviewee said, explaining that online options are less time-consuming and accessible for more employees.” (lines 363-365);
  • “One interviewee said while digitalization increased in speed, which affected organizational and treatment structures, all in all, the pandemic did not force the healthcare provider to introduce anything drastic. It was not the first pandemic they were handling. Ebola, swine fever, and the EHEC-crises prepared them, and emergency plans were in place to react swiftly and deliberately by redistributing resources and work. Another one also said that no structural changes are visible.” (lines 417-422)

Reviewer 2: 5. The digitization part is interesting.  It is not directly related to CSR, but could use more explanation (for example, are digital sessions with patients still desired?), and I think you could tie it directly to CSR more (discuss the impact on patients and employees of more digital sessions, the impact on the environment, etc.).  You might also expand the rest, emphasizing that the digitization that happened this time was a result of the specific nature of this crisis.

AUTHORS: We appreciate your input regarding the topic of digitalization and have expanded our explanation in this regard. For instance, we now write

  • “However, insurance companies only covered digital doctor-patient sessions for a certain amount of time. The healthcare sector still has a long way to go in terms of establishing well-rooted care options especially in Germany.” (lines 414-416);
  • “For instance, general CSR activities as one stream of action and digitalization as another should be pursued concurrently: Firstly, intentions towards both have to be formed and then planned in detail to actually change individual, group, and organizational behavior.” (lines 522-524);
  • “Digitalization, especially in crisis communication, matching other studies, and when exchanging safety information, was reportedly the factor getting the most sup-port through the pandemic, creating greater transparency. This can function as transfer cognition elements, leading to an exchange and facilitation of experiences. Although the initial plan to pursue CSR activities could not be fulfilled, the new experiences nonetheless had a positive impact. Using technology and being aware of the positive effects increases the opportunities for reaching goals. Through new regulations, the German government [59, 60] has paved the way for faster digitalization in the healthcare sector. The necessity for digital integration, communication, and knowledge transfer has become evident in the pandemic and was strongly fostered [61]. The created momentum can help to push the provision of medical services and the way medicine is practiced in new spheres. It remains to be seen how sustainable the changes in digitalization are, given that the presence of this specific pandemic was the driving factor.” (lines 533-545)

Reviewer 2: 6. You need to be careful to show the reader that with a sample of nine companies in Germany responding to one crisis, you don't think you've provided a definitive answer to the research question.  This relates in part to my comments about using the word "always," but if you think these findings generalize to other companies in other countries, other industries, and other crises, you need to call for future work to test whether this is so.  

AUTHORS: Thank you for these points. We adapted the manuscript to state the contribution of the research more clearly, answering the research question, and have refined our statement in the section about the study’s limitations.

The sections now read:

  • “The research that singles out the environmental aspects of CSR during crises is extremely limited [13], most research tries to establish an encompassing CSR approach including societal, organizational, and environmental aspects. However, this was rarely researched systematically, thus, the key research contribution of this study is to fill this gap.” (lines 46-50)
  • “As for practical implications, the study shows that most healthcare providers have similar experiences during the pandemic, no matter what size or field of specialty. Furthermore, the strain that the pandemic put on healthcare workers is immense, impacting their quality of life and well-being, satisfaction, and motivation. This knowledge can help support employees, make them more resilient, and provide options for finding a balance again to secure a strong and content workforce. Crisis management and crisis communication should take the larger picture into account, including coordinated strategic management.
  • The healthcare sector was put into a new spotlight due to the pandemic. This spotlight can be harnessed as a driving factor to foster sustainable changes and developments. Conducting further research on this topic can provide even more insights into the effect that crisis had on healthcare providers. These learned lessons can help to manage other crises, too, by means of transferring knowledge and developments.” (lines 584-596);
  • “As for CSR activities, the pandemic was a blessing and a curse at the same time. While the amount of waste and single-use plastics increased dramatically, online events and digitalization fostered sustainable development. This can be understood and facilitated making use of theories such as the CCAM which was used as theoreti-cal background for this study.
  • One major finding is the fact that many CSR activities were temporarily put on hold during the pandemic, showing that they are not yet fully incorporated into everyday work but rather considered as add-ons or nice-to-have activities. To fully integrate them into the business structures, CSR activities need to become an essential element of the business strategy that will not be disregarded once things get tough. The findings show that CSR might be a highly discussed topic within organizations but is not as firmly anchored within core business practices as it often seems. Hence, this re-search adds to the understanding of CSR practices in organizations and contributes to filling the research gap on how CSR is handled in the business context. The exact effects of the pandemic must be determined in the course of the coming months and years.” (lines 615-629)

Also, we added the need for further research to test the generalizability of the findings across sectors and countries. The section is now worded “The study’s main limitation is the number of participants which seems to be rather small at the first glance. However, given the fact that so many interviewees are working for a corporate group, they provided a vast array of knowledge. The interviewees were able to give insights into various facilities and with the fact that facilities of different sizes, as well as profit and non-profit organizations participated, the results are nonetheless representative. In addition, the results are similar to those generated in other countries. Chances are high that some aspects (changes in communication and digitalization) can be generalized across sectors and even countries, however, this hypothesis would need to be verified in future research.” (lines 575-583).

Reviewer 2: 7.  If I were to summarize the paper's findings, I would say that during the crisis, these companies put aside CSR efforts to focus on managing the crisis, and that some of the solutions they developed might continue to be useful after the crisis.  This seems reasonable, but not new.  I suggest that the revision should make more clear what the contribution is (c.f. my comment #3).

AUTHORS: Thanks for your assessment regarding the overall research contribution. We now give more information about this study’s contribution and hope the explanation makes strengthens this study’s value.

The section now reads “The study, therefore, contributes to filling the existing research gap, while at the same time giving specific insights into one sector by answering the research question: What CSR practices and activities have healthcare providers implemented in Germany and what effect does the pandemic have? This article makes two key contributions. First, it helps to gain a better theoretical and practical understanding of CSR activities in the face of crisis. Second, the empirical research will expand the knowledge of the distinct fields of single actions within CSR and the crisis due to the pandemic. While we know already that there are interrelations, this study will contribute to a holistic under-standing and guide for more effective encounters in the future. The article is structured as follows: the next section describes the methodology of the empirical study. The sub-sequent section reports the results obtained and analyzed, along with the synthesized categories. The last section closes with conclusions and future lines of research and practice.” (lines 173-184)

Reviewer 2: I hope these comments help.  Good luck!

AUTHORS: Thanks for your input. Your comments were more than helpful!

Reviewer 3 Report

he paper provides insightful evidence about what CSR practices and activities had healthcare providers implemented in Germany and what effect did the pandemic play. The research was well-design. Overall, the contents are nicely organized; it is quite easy to follow the background story and to digest the research findings. However, there are some points that still need improvement in order to enhance the overall quality and completeness of the paper. I encourage the authors to consider the comments given below and rigidly revise the paper as suggested to improve the overall quality and completeness of the research.

The contribution of the paper needs to be strengthened more in the introduction. The authors must clarify what is the key research contribution that the study provides. How this research expands the knowledge that we already know related this topic? Right now, it is not clearly stated in the paper.

On page 2, the review about the challenges that organizations faced during the covid-19 pandemic can still be expanded by including more research coverage about the management practices that are required to help employee cope effectively with stressors and uncertainties that organizations encountered during the covid-19 situation. In particular, I strongly recommend the authors to consider the 4 papers listed below as the additional references in the review about the organizational practices during the covid-19 pandemic. They provide important evidence about key management practices that are related to the key factors of the paper (e.g. communication, employee well-being, satisfaction, and motivation). Please incorporate them in the literature review and the discussion.

Influence of Transformational Leadership on Role Ambiguity and Work-Life Balance of Filipino University Employees During COVID-19: Does Employee Involvement Matter?, International Journal of Leadership in Education. https://doi.org/10.1080/13603124.2021.1882701

How Managerial Communication Reduces Perceived Job Insecurity of Flight Attendants During the COVID-19 Pandemic", Corporate Communications: an International Journal, 27(2), 368-387. https://doi.org/10.1108/CCIJ-07-2021-0080

Does the End Justify the Means? The Role of Organizational Communication among Work-from-Home Employees during the COVID-19 Pandemic. International journal of environmental research and public health, 18(8). doi:10.3390/ijerph18083933

Employee Engagement and Wellbeing in Times of COVID-19: A Proposal of the 5Cs Model. International journal of environmental research and public health, 18(10). doi:10.3390/ijerph18105470

Author Response

Reviewer 3: The paper provides insightful evidence about what CSR practices and activities had healthcare providers implemented in Germany and what effect did the pandemic play. The research was well-design. Overall, the contents are nicely organized; it is quite easy to follow the background story and to digest the research findings. However, there are some points that still need improvement in order to enhance the overall quality and completeness of the paper. I encourage the authors to consider the comments given below and rigidly revise the paper as suggested to improve the overall quality and completeness of the research.

AUTHORS: Thank you very much for your encouragement! We agree with all your comments and have accordingly rigidly revised our paper. We are positive that this process improved the overall quality and completeness of the research.

Reviewer 3: The contribution of the paper needs to be strengthened more in the introduction. The authors must clarify what is the key research contribution that the study provides. How this research expands the knowledge that we already know related this topic? Right now, it is not clearly stated in the paper.

AUTHORS: Thank you very much for this input. We have enhanced the sections about the study’s contribution by including literature supporting the clarification of the research gap and by clearly stating the study’s intention.

The sections now read:

  • “The research that singles out the environmental aspects of CSR during crises is extremely limited [13], most research tries to establish an encompassing CSR approach including societal, organizational, and environmental aspects. However, this was rarely researched systematically, thus, the key research contribution of this study is to fill this gap.” (lines 46-50)
  • “The study, therefore, contributes to filling the existing research gap, while at the same time giving specific insights into one sector by answering the research question: What CSR practices and activities have healthcare providers implemented in Germany and what effect does the pandemic have? This article makes two key contributions. First, it helps to gain a better theoretical and practical understanding of CSR activities in the face of crisis. Second, the empirical research will expand the knowledge of the distinct fields of single actions within CSR and the crisis due to the pandemic. While we know already that there are interrelations, this study will contribute to a holistic under-standing and guide for more effective encounters in the future. The article is structured as follows: the next section describes the methodology of the empirical study. The sub-sequent section reports the results obtained and analyzed, along with the synthesized categories. The last section closes with conclusions and future lines of research and practice.” (lines 173ff).

Reviewer 3: On page 2, the review about the challenges that organizations faced during the covid-19 pandemic can still be expanded by including more research coverage about the management practices that are required to help employee cope effectively with stressors and uncertainties that organizations encountered during the covid-19 situation. In particular, I strongly recommend the authors to consider the 4 papers listed below as the additional references in the review about the organizational practices during the covid-19 pandemic. They provide important evidence about key management practices that are related to the key factors of the paper (e.g. communication, employee well-being, satisfaction, and motivation). Please incorporate them in the literature review and the discussion.

AUTHORS: Thank you very much for making us aware of this. We included all of the four papers (see the corresponding sections through the revision mode) and in addition also incorporated six more into the introduction and the discussion, concretely, those added ones are the following:

  • Marie Lauesen, L. CSR in the aftermath of the financial crisis. Soc. Responsib. J., 2013, 9(4), 641-663. https://doi.org/10.1108/SRJ-11-2012-0140
  • Souto, B. F.-F. Crisis and corporate social responsibility: Threat or opportunity?. Int. J. Econ. Sci. Applied Res., 2009, 2(1), 36-50.
  • Bae, K.-H.; El Ghoul, S.; Gong, Z. J. & Guedhami, O. Does CSR matter in times of crisis? Evidence from the COVID-19 pandemic, J. Corp. Finance, 2021, 67. https://doi.org/10.1016/j.jcorpfin.2020.101876
  • Ham, C.-D. & Kim, J. The effects of CSR communication in corporate crises: Examining the role of dispositional and situational CSR skepticism in context. Public Relat. Rev. 2020, 46(2). https://doi.org/10.1016/j.pubrev.2019.05.013
  • Zhou, Z. & Ki, E.-J. Exploring the role of CSR fit and the length of CSR involvement in routine business and corporate crises settings, Public Relat. Rev., 2018, 44(1), 75-83. https://doi.org/10.1016/j.pubrev.2017.11.004
  • Seles, B. M. R. P.; Lopes de Sousa Jabbour, A. B., Chiappetta Jabbour, C. J. & Jugend, D. In sickness and in health, in poverty and in wealth?: Economic crises and CSR change management in difficult times. J. Organ. Change Manag., 2018, 31(1), 4-25, https://doi.org/10.1108/JOCM-05-2017-0159

Reviewer 3:

  • Influence of Transformational Leadership on Role Ambiguity and Work-Life Balance of Filipino University Employees During COVID-19: Does Employee Involvement Matter?, International Journal of Leadership in Education. https://doi.org/10.1080/13603124.2021.1882701
  • How Managerial Communication Reduces Perceived Job Insecurity of Flight Attendants During the COVID-19 Pandemic", Corporate Communications: an International Journal, 27(2), 368-387. https://doi.org/10.1108/CCIJ-07-2021-0080
  • Does the End Justify the Means? The Role of Organizational Communication among Work-from-Home Employees during the COVID-19 Pandemic. International journal of environmental research and public health, 18(8). doi:10.3390/ijerph18083933
  • Employee Engagement and Wellbeing in Times of COVID-19: A Proposal of the 5Cs Model. International journal of environmental research and public health, 18(10). doi:10.3390/ijerph18105470

AUTHORS: Thank you very much for bringing these great fitting papers to our attention. We included them into the introduction and the discussion.

Round 2

Reviewer 2 Report

Thank you for your hard work on this paper.  I see significant improvements on many points.  I have one remaining concern.

CSR has been studied intensively, exhaustively, for decades.  The study in its current form does not just fail to recognize that work, it dismisses it out of hand.  In the abstract, you say, "However, CSR has been scarcely researched systematically."  This is the only grammatical error I noticed in this revision, but more importantly, it is simply not true.    In the introduction (lines 34-37), you open with, "It is well known that Corporate social responsibility (CSR) or sustainability practices within corporations are a key component in all sectors, including the healthcare system. Corporations and organizations are becoming increasingly aware of the significance of CSR, focusing on using their available resources sustainably while governments simultaneously started to regulate the use of these resources."  You offer no evidence to support these claims, other than a general "it is well-known."  Scholars in the field do know such things because we have done the research, published the research, read the research, and cited the research.  You should cite the research here, and at some point in the study you should offer a brief review of the relevant CSR literature on which this study builds.  You need to show us, maybe in the conclusions, how this article's contributions fit into the CSR literature.  Also, I see in your response to my previous comment on this topic that most of the CSR literature looks at all three areas (social, environmental, and organizational), and that is true, but (1) the current study is about CSR, so those findings should be relevant, (2) there is a subfield that refers to "environmental CSR" (you can search for that in Google Scholar, (3) multiple studies of CSR break down and report on differences among the three areas.  

This is not a large change, as changes to articles go, but it is important.

Author Response

REVIEWER: Does the introduction provide sufficient background and include all relevant references? Must be improved:

AUTHORS: Thank you very much for this feedback and the opportunity to improve our manuscript. We have revised to introduction and added more background information on the study-relevant CSR literature and on environmental CSR, which also nicely ties in with your last remark. Please the version of our manuscript with track changes accordingly.

REVIEWER: Are all the cited references relevant to the research? & Are the conclusions supported by the results? Can be improved:

AUTHORS: Thank you for these comments. We checked the cited references again do think that all are relevant. The extensive revision, especially of the introduction, should clarify the importance of the literature. We have also adjusted the conclusion to demonstrate even more and clearer how the results support the conclusion. Please see our revisions highlighted in track changes.

REVIEWER: CSR has been studied intensively, exhaustively, for decades.  The study in its current form does not just fail to recognize that work, it dismisses it out of hand.  In the abstract, you say, "However, CSR has been scarcely researched systematically."  This is the only grammatical error I noticed in this revision, but more importantly, it is simply not true.  

OUR RESPONSE: We are very sorry for this error and have corrected it. The sentence now reads “However, theory-driven CSR research within the healthcare sector is scarce“ (See the abstract).

REVIEWER:   In the introduction (lines 34-37), you open with, "It is well known that Corporate social responsibility (CSR) or sustainability practices within corporations are a key component in all sectors, including the healthcare system. Corporations and organizations are becoming increasingly aware of the significance of CSR, focusing on using their available resources sustainably while governments simultaneously started to regulate the use of these resources."  

OUR RESPONSE: We apologize for incompleteness and have replaced the two sentences by the following paragraph “Corporate social responsibility (CSR) is a concept that is based on a company’s will to go beyond its obligations to maximize social, environmental (with a strong focus on climate), economic, and ethical welfare [1-6]. Definitions on the topic strongly vary and so do the terms frequently used in this context. Apart from CSR, “corporate sustainability” [7] or “triple bottom line” [8] are often utilized. All previous researchers seem to agree that the following elements need to be included in a definition of the concept:

The three main areas of engagement are society, environment, and business – whereby society usually refers to doing something good for society; involvement in environmental activities ranges from supporting environmental and climate initiatives by only sourcing local or fair-trade products, to re-evaluating and changing a company’s statutes to become more (i.e. energy) efficient; and business denotes economic decisions, such as becoming an attractive employer through extending additional benefits to employees or by making processes more efficient, all leading to increased satisfaction and sustainability. Quite often, approaches to support and strengthen human rights or to take the opinions of customers into account explicitly mention these three elements to emphasize their importance [9-11]. Complementing these are the legal and ethical spheres [12], elements of which are found in all three areas. In addition, fundamental philanthropical ideas can often be found in various CSR activities [13].

External social, economic, and environmental factors play into the decision-making processes within a company. Such factors can include for example regulators, employees, shareholders, customers, activists, researchers, or any other form of stakeholder [14].

Stakeholders legitimize the company’s CSR activities. This is often shown in terms of how much support a company gets for its activities, i.e. donations, time employees spend on helping out and engaging [15,16], and how much stakeholders trust that the company’s image is befitting of its actions [17,18].

CSR is rooted deeply within business practices and applied activities:[…]“ (See lines 29ff).

REVIEWER: You offer no evidence to support these claims, other than a general "it is well-known."  Scholars in the field do know such things because we have done the research, published the research, read the research, and cited the research.  You should cite the research here, and at some point in the study you should offer a brief review of the relevant CSR literature on which this study builds.  

OUR RESPONSE: We are regret not providing references and have done so now. Please see the revised manuscript with tracked changes.

REVIEWER: You need to show us, maybe in the conclusions, how this article's contributions fit into the CSR literature.

OUR RESPONSE: Thank you very much for this advice. We have accordingly revised the conclusion and one exemplary sentence now reads “The article's contributions into the CSR literature is the use of an innovative theory and its empirical exploration during times of crisis i.e. corona pandemic on CSR activities of healthcare providers.“ (See lines 872ff).

REVIEWER: Also, I see in your response to my previous comment on this topic that most of the CSR literature looks at all three areas (social, environmental, and organizational), and that is true, but (1) the current study is about CSR, so those findings should be relevant, (2) there is a subfield that refers to "environmental CSR" (you can search for that in Google Scholar, (3) multiple studies of CSR break down and report on differences among the three areas.

OUR RESPONSE: Thank you for raising these points. We have revised and enriched the introduction, giving evidence to support the claims, citing the relevant research for the study, and adding more input on environmental CSR, while connecting it our own research (please see highlighted in the text). Throughout the text, we have tried our very best to clarify the article’s contributions to the existing literature and we sincerely hope that this is satisfactory.

Here are some examples thereof:

“Apart from the COVID pandemic, climate change is a relevant factor that negatively influences the health of individuals and societies [33,42,43]. Global warming is one aspect that strongly impacts people all around the world [44]. Having many pointing the finger at companies rather than individuals for having a negative influence on the environment and society, managers all over the world implement corporate social responsibility or sustainability practices within their corporations. In Germany, one of the sectors with no specific regulations regarding CSR is the healthcare sector. Healthcare providers are not required by law to be sustainable although hospitals especially consume and waste an enormous amount of energy because providers very often do not evaluate savings potential [45]. The pandemic has a major impact on environmental efforts, the next few years will show the specific fallouts. This study enriches the research by giving first insights into the effects of the pandemic on environmental CSR activities, building on previous research.” (See lines 98ff).

“In the field of environmental CSR, research has already reported interesting general aspects: For example, well-written and detailed CSR disclosure contributes to transparency by displaying the environmental and social impact precisely [46]. Fully committing to environmental corporate social responsibility business practices “could avoid public skepticism that would negate the organization’s effort in strengthening customer loyalty” [47], (p. 710). Babiak and Terndafilova looked at sports organizations’ motives to adopt environmental practices and found that “environmental practices in professional sport are driven by two important considerations – the desire to achieve legitimacy and the strategic or competitive advantages that these types of activities might provide.” [48] (p. 22). And with a specific focus on the healthcare sector, a study by Xu et al. recently looked at the pro-environmental behavior of healthcare staff connected to their institutions’ attitudes and actions regarding energy efficiency [49]. They found out that the staff learns from what they experience within their institution, its attitude and behavior are thus highly relevant. Employees consequently play an important role and their engagement is one of this study’s main foci.” (see lines 156ff).

“In a recent study Möhring, Naumann, Reifenscheid et al. [60] showed that not enough research emphasis has been put on the severe social consequences brought by the pandemic. Neither in terms of how individuals behave among each other nor what is socially acceptable or preferred by society. This is particularly important concerning the stipulations on adhering to physical distancing and hygiene rules i.e. minimizing personal contact to stop the virus from spreading [68]. To promote organizational health and well-being, a model for employee engagement focusing on conciliation, cultivation, confidence, compensation, and communication was proposed [69], combining many of the aspects other researchers have looked at in their work. Linking to this previous research, one of the key research contributions of this study is to provide new healthcare-sector-specific knowledge, expanding the existing research.

Previously, little theoretical work was done in the area of CSR (e.g., [70]) with regard to experiences and concrete behaviors and there are a few different models being applied to social accounting [71]. Therefore, a well-developed model of health behavior change is applied to CSR, which will be described in the following and used for the empirical work of this study (e.g., [69]).” (See lines 220).

“While the initial study’s focus was on one particular country, namely Germany, and its healthcare providers’ CSR activities, the pandemic’s influence on these activities is far reaching across sectors and country boundaries. Therefore, the research needs to be enriched by questions specific to places from all over the world, identifying how the pandemic was handled to understand how crisis management works and what lessons can be learned for the future. A previous study in Pakistan found that the CSR engagement of a hospital, including the employees, is imperative for promoting workplace innovation and psychological safety [74]. Accordingly, the current study aims to provide in-depth insight into the effects, changes, and challenges of the COVID-19 pandemic and the motivation, satisfaction, and well-being of the employees. As shown above, the research on CSR during crises is very limited, the COVID-19 pandemic provides a crucial opportunity to explore this field of research. The study, therefore, contributes to filling the existing research gap, while at the same time giving specific insights into one sector by answering the research question: What CSR practices and activities have healthcare providers implemented in Germany and what effect does the pandemic have? This article makes two key contributions. First, it helps to gain a better theoretical and practical understanding of CSR activities in the face of a crisis. Second, the empirical research will expand the knowledge of the distinct fields of single actions within CSR and the crisis due to the pandemic. While we know already that there are interrelations, this study will contribute to a holistic understanding and guide for more effective encounters in the future. The article is structured as follows: the next section describes the methodology of the empirical study. The subsequent section reports the results obtained and analyzed, along with the synthesized categories. The last section closes with conclusions and future lines of research and practice.” (lines 377).

“This study looked into the challenges and opportunities the German healthcare sector faced during the second wave of the Corona pandemic, with a special focus on CSR activities, based on the Compensatory Carry-over Action Model (CCAM). The results offer insights into changes, challenges, and opportunities care providers are currently confronted with. The different approaches to handling them show the opportunities to learn from the pandemic, change behavioral patterns, and open up a discourse about the way forward. Especially through the lens of CSR, the study shows how gravely the pandemic impacted different CSR activity efforts throughout the entire sector but also beyond CSR the work in general in face of crisis, and options of CSR to help to cope with crises. to continue the path towards a more sustainable healthcare sector, CSR activities are either even higher on the priority list or the first projects to be stopped or abandoned during a crisis. This results in the previous actions and behaviors being either turned back or stopped. CSR drifted out of corporate and public focus, and organizations fell behind on their commitments. But at the same time, CSR was found to facilitate sustainability of quick coping actions.“ (See lines 703ff)

“Hence, this research adds to the understanding of CSR practices in organizations and contributes to filling the research gap on how CSR is handled in the business context. The exact effects of the pandemic must be determined in the course of the coming months and years.” (See lines 883ff).

We really hope the reviewer agrees with us that we now cite the relevant previous research and the relevant CSR literature on which our study builds. We adapted the conclusions, how this article's contributions fit into the CSR literature and hope the reviewer is satisfied with this.

Reviewer 3 Report

The authors did a very good job in handling all comments provided in the previous evaluation. Now the quality of the paper is adequately improved. There is no further comment for the authors from my side.

Author Response

Thank you very much for your appreciation.